# Mechanisms and Management of Thyroid Disease and Atrial Fibrillation: Impact of Atrial Electrical Remodeling and Cardiac Fibrosis

**DOI:** 10.3390/cells11244047

**Published:** 2022-12-14

**Authors:** Abhijit Takawale, Martin Aguilar, Yasmina Bouchrit, Roddy Hiram

**Affiliations:** 1Research Center, Montreal Heart Institute, Montréal, QC H1T 1C8, Canada; 2Department of Pharmacology and Therapeutics, McGill University, Montreal, QC H3A 0G4, Canada; 3Department of Medicine, Faculty of Medicine, University of Montreal, Montreal, QC H3T 1J4, Canada

**Keywords:** thyroid disease, atrial fibrillation, electrophysiology, atrial fibrosis, inflammation

## Abstract

Atrial fibrillation (AF) is the most common cardiac arrhythmia associated with increased cardiovascular morbidity and mortality. The pathophysiology of AF is characterized by electrical and structural remodeling occurring in the atrial myocardium. As a source of production of various hormones such as angiotensin-2, calcitonin, and atrial natriuretic peptide, the atria are a target for endocrine regulation. Studies have shown that disorders associated with endocrine dysregulation are potential underlying causes of AF. The thyroid gland is an endocrine organ that secretes three hormones: triiodothyronine (T3), thyroxine (T4) and calcitonin. Thyroid dysregulation affects the cardiovascular system. Although there is a well-established relationship between thyroid disease (especially hyperthyroidism) and AF, the underlying biochemical mechanisms leading to atrial fibrosis and atrial arrhythmias are poorly understood in thyrotoxicosis. Various animal models and cellular studies demonstrated that thyroid hormones are involved in promoting AF substrate. This review explores the recent clinical and experimental evidence of the association between thyroid disease and AF. We highlight the current knowledge on the potential mechanisms underlying the pathophysiological impact of thyroid hormones T3 and T4 dysregulation, in the development of the atrial arrhythmogenic substrate. Finally, we review the available therapeutic strategies to treat AF in the context of thyroid disease.

## 1. Introduction

The thyroid gland is an endocrine organ responsible for the secretion of triiodothyronine (T3), thyroxine (T4) and calcitonin (CT) [1]. Thyroid hormones regulate a wide range of physiological functions, including the basal metabolic rate, growth, and maturation, as well as modulating the autonomic nervous and respiratory systems [1]. The thyroid and the heart have common embryonic origin, and the heart is an important target-organ for thyroid hormones [1]. The British physician C. Parry was the first to report on an association between thyroid disease, specifically thyroid enlargement, and increased heart rate in 1825 [2,3].

Atrial fibrillation (AF) is the most common sustained cardiac arrhythmia in the general population and is associated with significant cardiovascular morbidity and mortality [4]. There is epidemiological evidence of the association between thyroid disease and AF [4,5,6]. Numerous clinical studies have reported that hyperthyroidism is associated with an increased risk of AF [4,5,6]. Thyroid ablation reduces the incidence of AF [7] in patients with thyroid disease such as Graves’ disease [8], nodular goiter [9], and hyperthyroidism [10]. Cardiac disorders associated with thyroid disease are characterized by electrophysiological changes such as increased P wave duration, pulmonary vein ectopy, and sinoatrial (SA) node automaticity [11]. Altered regulation of thyroid gland can potentiate AF inducibility, particularly in older individuals in which lower thyroid stimulating hormone (TSH) has been shown to be associated with increased AF occurrence [12]. Similarly, low TSH levels, or high T3/T4 circulating levels have been associated with AF recurrence after catheter ablation [13,14]. Here, we discuss the impact of thyroid disorder on the development of atrial fibrosis, which we suspect is one of the main contributors explaining the risk of AF in thyrotoxicosis.

In this review, we focus on the experimental and clinical evidence linking atrial fibrosis, AF and thyroid diseases; we summarize the current therapeutic approaches for the management of AF patients with thyroid disease; and we discuss future directions for the prevention and treatment of thyroid disease-associated AF.

## 2. Clinical Evidence of Thyroid Disease and AF

There is a well-established epidemiological association between thyroid disease and AF such that thyroid function testing is recommended in all patients with new-onset AF [15]. Overt hyperthyroidism (low TSH, elevated T4) has the strongest association with AF. It increases the risk of AF up to 3-fold [12,16], but is relatively uncommon [17]. More recently, subclinical hyperthyroidism (low TSH with normal T3/T4), often regarded as a benign entity, has also been reported to carry an increased risk of AF (hazard ratio in the range of 1.5) [4,18,19,20,21,22]. Similarly, a large phenome-wide association study found TSH levels within the normal range to be inversely related to AF (odds ratio 0.91; 95% confidence interval, 0.83–0.99; *p* = 0.04), supporting the notion that even subclinical increases in thyroid function are associated with AF [23]. Subclinical hyperthyroidism is present in up to 13% of patients with new-onset AF such that it may represent an underappreciated reversible cause or contributing factor in a sizeable number of patients [24].

Conversely, the association between hypothyroidism, even clinical (high TSH, low T4) is much less robust. In fact, several large clinical studies found no residual correlation between clinical or subclinical hypothyroidism and AF after adjusting for covariates [17,22]. From a clinical standpoint, hypothyroidism is associated with a range of cardiovascular conditions which often co-exist with AF, such as hypertension and diastolic dysfunction [25,26]. Many patients may therefore have concomitant hypothyroidism and AF, but there is no good evidence that restoring the euthyroid state directly impacts the natural history of AF.

The mechanisms by which hyperthyroidism leads to AF in humans are poorly understood. The relatively small body of animal experimental data is reviewed below. Even less is known about the relationship between thyroid disease and atrial electrophysiology in humans. Hyperthyroid patients have increased heart rates, more atrial ectopic activity (atrial premature contractions) and decreased heart rate variability, which often normalize with restoration of the euthyroid state [27,28] (Table 1).

The significance of these observations in relation to AF are unclear and there is no published systematic evaluation of atrial electrophysiological changes in hyperthyroid humans.

## 3. Mechanisms of Thyroid Disease-Induced Arrhythmogenic Remodeling

### 3.1. Thyroid Hormones and Cardiomyocytes

The mammalian heart has limited intracellular deiodinase activity, and only T3 but not T4 is transported into cardiac myocytes via protein transporters [29,30]. Thyroid hormones exert genomic actions and non-genomic actions to modulate cardiac function. Genomic actions are mediated primarily through T3 interaction with thyroid receptors (TR; TRα and TRβ) expressed on the cardiomyocytes (CM) cell membrane [31,32]. The TRα splice variants includes the T3-binding isoform TRα1 and the 3′-splice variant TRα2 which does not bind to T3. The TRβ isoforms has 3 spice variants (TRβ1, TRβ2, TRβ3) [33,34]. These genomic actions on cardiac myocyte further involves transcriptional changes to alter myocyte functions. Non-genomic actions have a much more rapid onset of action as they do not require transcription and appear be involved in the modulation of cell membrane ion channels [3] (Figure 1).

### 3.2. Thyroid Hormones and Atrial Electrical Remodeling

In isolated ventricular CM, T3 positively upregulates myosin heavy chain, alpha isoform (MHC-α) and sarcoendoplasmic reticulum calcium (Ca^2+^) transport adenosine triphosphatase (ATPase) (SERCA) while negatively regulating phospolamban (PLN) and (MHC-β), contributing to increased CM contractility [24,25]. In a rat model of hypothyroidism, seven days of T3 treatment (7 μg/day by constant infusion) has been shown to cause significant changes on contractile proteins levels, including increased PLN (in LA and LV) and decreased SERCA (in LV) with atria showing greater changes compared to ventricles [35]. A rat model of hypothyroidism induced by methimazole showed upregulation in ion channel protein content of Kv1.5, Kv7.1, and Cav1.2 associated with structural remodeling that caused shortening of atrial effective refractory period (AERP) and increased AF susceptibility [36].

Acute exposure of T3 to cat atrial CM caused increased contractility and Ca^2+^-mediated triggered activity. These changes were mediated by activation of the inward sodium (Na^+^; INa) current and stimulation of Na^+^-Ca^2+^ exchanger current (INCX) [37,38]. There is evidence to suggest that T3 directly interacts with CM membrane by slowing inactivation of sodium channels [39], and decreasing expression of atrial L-type Ca^2+^ channels in hyperthyroidism, shortening of action potential duration (APD) potentially due to inhibition of cyclic adenosine monophosphate (cAMP) response element binding protein and its nuclear phosphorylation [40] (Figure 1).

These interactions between T3 and the regulation of CM ion channels suggest a potential role of thyroid hormones in modulating atrial electrophysiological properties.

### 3.3. Thyroid Hormones, Atrial Fibrosis and Atrial Structural Remodeling

Atrial fibrosis is an important component of AF pathobiology [41,42]. Cardiac fibroblasts (FB) are responsible for secreting and maintaining the extracellular matrix. Interestingly, FB express only a tenth of the number of thyroid receptors per cell in comparison with cardiac myocytes [3]. FB predominantly express TRβ1 receptors, whereas TRα1 is the predominant receptor on CM [43]. It has been shown that L-thyroxine administration decreases ventricular FB collagen I expression in a time dependent manner [44,45]. Additionally, increases in T3 and T4 have been shown to significantly increase metabolism in particular with increased glucuronidation of T3 and T4 in FB compared to CM [46]. Thyroid hormone depletion caused enhanced cardiac fibroblast proliferation, enhanced DNA synthesis, and increased collagen mRNA [43]. These events may contribute to the development of cardiac fibrosis. In fact, supplementation in thyroid hormones was reported to cause increased matrix metalloproteinase (MMP) activity, while cardiac pathologies such as ischemia and hypertrophic cardiomyopathy were associated with increased collagen accumulation in hypothyroidism [47,48,49]. These experimental observations suggest that strategies promoting optimal physiological levels of thyroid hormones could help to decrease myocardial fibrosis [48] (Figure 1 and Figure 2).

This indirect evidence suggests a potential role of thyroid hormones in cardiac fibrosis development. Further studies are required to evaluate the potential role of thyroid hormones in mediating atrial FB malfunction and arrhythmogenic atrial structural remodeling.

### 3.4. Thyroid Hormones, Cardiac Metabolism, and Mitochondrial Remodeling

The heart is a highly metabolically active organ. Oxidative stress and mitochondrial stress can negatively affect atrial myocardial function and contribute to AF [50,51,52]. Clinical and experimental studies have showed that altered thyroid states are associated with increased cardiac oxidative stress that is mediated through cell membrane damages, and unsaturated phospholipids accumulation [53,54,55,56]. Moreover, elevated levels of thyroid hormones induce a hypermetabolic state [57], which alters the activity of mitochondrial respiratory complexes, promoting the generation of highly reactive superoxide (O^2−^) ions [58]. In the mitochondria, various T3 binding sites, such as the adenine nuclear transporter (ANT) have been identified, suggesting a potential effect of thyroid hormones on mitochondrial function [57]. Hyperthyroid models showed damaged cardiac mitochondrial ultrastructure, increased mitochondrial ATP activity, and oxidative stress [59,60,61] (Figure 1). In an ischemia-reperfusion model, T3 supplementation has been shown to improve mitochondrial functions associated with increased myocardial ATP levels and increased aerobic metabolism [62,63].

### 3.5. Thyroid Hormones and Cardiac Inflammation

Interaction of highly reactive superoxide ions and nitric oxide (NO) generate peroxynitrite, that activate immune response and cytotoxicity [56]. Nitric oxide synthase (NOS) catalyzes synthesis of NO and its inducible isoforms which are essential signaling molecules in cardiac physiology [58]. Studies have shown that hyperthyroidism promotes NOS activity in the myocardium, and nicotinamide adenine dinucleotide phosphate (NADPH) oxidase-mediated cardiac hypertrophy [58]. It is well accepted that inflammation is also an important contributor of AF incidence [42]. Thyroid hormones activate various inflammatory pathways, including tumor necrosis factor alpha (TNF-α), interleukin (IL)-6, IL-1, interferon gamma (IFN-γ), potentially due to activated free radicals, and peroxynititre [64] (Figure 1 and Figure 2). Recent data suggest that enhancement of such inflammatory signaling in the atrium is implicated in the development of arrhythmogenic substrate promoting the occurrence and recurrence AF [65]. However, the detailed mechanisms of the association between inflammation, thyroid abnormalities and atrial fibrillation remain unclear.

## 4. Experimental Studies of Arrhythmias in Thyroid Disorders

The association between thyroid disease and AF is accepted, however few experimental studies have been reported. Little is known about the precise mechanisms responsible for the causality link between AF and thyroid disorders. Further understanding of the pathophysiology of thyroid disease association with AF may provide novel therapeutic approaches to the clinical management of these patients. The following section provides an original review of the basic research linking thyroid disease and AF. Reported experimental studies of thyroid disease and AF include molecular and cellular approaches, murine, rats, rabbits, dogs, and primates’ models.

### 4.1. Molecular and Cellular Studies of Thyroid Hormone and AF

In a cellular model [66], Chen YC et al. have shown that hyperthyroid-like conditions led to pulmonary veins (PV) electrical remodeling characterized by enhanced automaticity and triggered activity. Given the prominent role of the PV in AF, this may be an important mechanism through which hyperthyroidism contributes to the development of AF [66]. In another study, hyperthyroidism-induced molecular changes were investigated using atrial CMs isolated from hyperthyroid rodents [40]. Expression of the L-type Ca^2+^ channel was decreased, and APD was shortened [40]. These in vitro data suggest that hyperthyroidism may induce dysregulated expression and activity of key ion channels and proteins responsible for: perturbation of atrial electrical activity, abnormal conduction velocity and development of AF [67] (Figure 2 and Table 2).

### 4.2. Murine Models of Arrhythmias in Thyroid Disease

In a murine model of hyperthyroidism, whole-cell patch-clamp was used to determine the ADP and ionic conductance in left and right atrial CM from control vs. hyperthyroid mice. The investigators observed that hyperthyroidism led to significant APD shortening and increased delayed rectifier K^+^ current, more so in the right atria (RA) [68]. It is known that these electrical remodeling can facilitate the occurrence of re-entry circuits which promote the atrial arrhythmias’ substrate [42]. Another study aiming to evaluate the role of thyroid hormones on cardiac atria has found that hypo- and hyperthyroid mice, respectively showed 25% decrease and 40% increase in connexin (Cx) 40 (Cx40) mRNA expression [69] (Figure 2 and Table 2). These results reveal that atrial Cx40 expression and specific potassium channels are influenced by thyroid hormone, suggesting a mechanistic link between thyroid disease and proarrhythmic electrical changes in the form of altered conduction and abnormal repolarization. 

Furthermore, a recent translational study investigated the role of calcitonin (CT), a paracrine hormone secreted by the thyroid gland, in the pathophysiology of AF. The authors reported that atria-specific knockdown of CT promotes atrial fibrosis and AF while its overexpression prevents the development of atrial arrhythmogenic substrate, suggesting an alternative pathway through which the thyroid may be involved in AF [70]. In this research article published in Nature, the authors reported that compared to control mice, overexpression of CT was associated with decreased FB proliferation and decreased mRNA levels of Acta2 (Actin Alpha 2, Smooth Muscle) [70].

### 4.3. Rat Models of Thyroid Disease Associated with AF

The association between AF and thyroid disease has also been studied in rats. Hypothyroid and hyperthyroid animals showed cardiac structural and functional abnormalities compared to euthyroid rats. Heart rate, sinus node recovery time, atrial effective refractory period, and atrioventricular conduction time were affected by thyroid disease. Researchers revealed that hypothyroidism significantly increased LA fibrosis [71]. Moreover, AF incidence and AF duration were higher in rats with hyperthyroidism and/or hypothyroidism compared to euthyroid animals [71] (Figure 2 and Table 2). In another rat model of thyroidectomy-induced hypothyroidism, it was reported that, compared to normal rats, hypothyroid animals had significantly lower heart rate, shortened atrial effective refractory period and prolonged sinus node recovery time as well as increased bi-atrial volume and increased AF inducibility and duration [36]. Atrial fibrosis and transforming growth factor bêta (TGFβ) expression were also increased in hypothyroid rats compared to control [36]. The same study also found upregulation of Kv1.5, Kv4.2, Kv4.3, Kv7.1, Cav1.2, and downregulation of Kir3.1 and Kir3.4 in hypothyroid rats [36] (Table 2).

Altogether, these structural and electrophysiological remodeling induced by hypothyroidism may contribute to the development of atrial arrhythmias, including AF.

### 4.4. Rabbit Model of the Association between Thyroid Disease and AF

Hyperthyroidism was induced in rabbits using daily injections of T4. The authors observed that increased thyroid hormone and β2-adrenergic-receptor activation played a synergistic arrhythmogenic effect [71]. Increased thyroid hormone secretion was associated with enhanced occurrence of atrial tachycardia. Results reported by Li H and colleagues in 2014 were consistent with Arnsdorf’s and Childers’ study who noticed, since 1970, that rabbit model of hyperthyroidism involving daily injections of 250 μg/kg of l-thyroxine during seven days, was associated with increased heart rate, shorter atrial ERP, and increased atrial arrhythmias including sinus tachycardia, atrial tachycardia, and atrial fibrillation [72,73] (Figure 2 and Table 2).
cells-11-04047-t002_Table 2Table 2Experimental models of thyroid diseases associated with AF. Review of articles investigating the incidence of atrial fibrillation in pre-clinical models of thyroids disorders, including experiments performed in vitro (molecular and cellular approaches) and in vivo (in mice, rats, rabbits, dogs, and primates). Abbreviations: AF = atrial fibrillation; APD = action potential duration; Ca^2+^ = calcium; CT-KD = calcitonin knockdown; CREB = cyclic AMP response element (CRE)-binding protein; Cx = connexin; ECG = electrocardiogram; ERP = effective refractory period; K^+^ = potassium.
ThyroidDiseaseAF SubstrateAF IncidenceECG PhenotypeRefs.Molecule & CellHyperthyroidismShortened Atrial APDElectrical remodelingNot tested[66]HyperthyroidismShortened Atrial APDDecreased Atrial L-type Ca^2+^ channelInhibition of CREB activityNot tested[40]MouseHyperthyroidismDecreased APDIncreased delayed K^+^ rectifier currentNot tested[68]Hypothyroidism andHyperthyroidismCx40 deregulationAbnormal atrial conduction velocityProlonged:RR interval,P-wave durationPR segment.[69]Calcitonin knock-downFibroblast proliferation, Enhanced Atrial fibrosisIncreased cAMP productionAF incidence:72% in CT-KD[70]RatHypothyroidism andHyperthyroidismMyocardial fibrosisAtrial structural & functional remodelingAtrial ERP changesAF incidence:78% in hypothyroid67% in hyperthyroid11% in euthyroid[44]HypothyroidismAtrial fibrosis, Atrial stretchIon channels alterationAF incidence:~90% in hypothyroid~17% in euthyroid[8]RabbitHyperthyroidismDecreased Atrial ERPLowering of atrial nultiple-response thresholdAF incidence:11% in hyperthyroid[73]HyperthyroidismEnhanced arrhythmogenic effect ofβ2-adrenergic receptors activationAF incidence:~15% in hyperthyroid[72]DogHyperthyroidismSlowed atrial conduction velocityProlonged P-R intervalsAF incidence:81% in hyperthyroid[74]HypothyroidismLow atrial voltage zonesAF incidence:7% in hypothyroid[75]PrimateHyperthyroidismIncreased β2-adrenergic receptors activityIncreased heart rate[76]

### 4.5. Dog Model of Hyperthyroidism Associated with AF

In 1956, Dr. Phillip E. Leveque, Ph.D. proposed a dog model of thyrotoxicosis generated by daily feeding with 3 g of thyroid-extract powder, mixed with dog’s food. Animals were tested daily to determine the establishment of thyrotoxicosis, and cardiac electrophysiological studies were performed to evaluate AF incidence. Thyroid hormone administration was associated with occurrence of anorexia, diarrhea, and tachycardia. Induced hyperthyroidism provoked increased sensitivity to acetylcholine. Hyperthhyroidism-associated tachyarrhythmias included severe conduction block, prolonged P-R intervals, and frequent episodes of AF [74]. In 1996, a veterinaty report from Gerritsen R.J. and collaborators evaluating AF incidence in 401 dogs, concluded that primary hypothyroidism was significantly higher in dogs with AF compared to control animals [75] (Table 2). Together, available data on AF incidence in dog with thyroid disease suggest that homeostatic levels of thyroid hormones might contribute to prevent development of cardiac arrhythmias.

### 4.6. Primate Model of Cardiopathy Associated with Thyroid Disease

In 1997, the contribution of β2-adrenergic receptors in the pathophysiology of thyroid disease-induced AF has also been demonstrated by Brian D. Hoit and collaborators, in a primate model using baboons which food was supplemented with T4 [76]. The authors have shown that hyperthyroidism leads to significantly increased myocardial expression of β2-adrenergic receptors compared to control animals. This phenomenon was associated with ventricular tachycardia [76] (Table 2). Little is known about the impact of thyroid disease on atrial function in primates however, the recent evidence from smaller mammals, including mice, as described above, is consistent to support the hypothesis that thyroid hormones’ regulation is important to prevent and/or control atrial arrhythmogenicity.

Experimental models are fundamental to better understand the pathophysiological mechanisms governing the association between AF and thyroid disease. More investigations are required to understand whether hyperthyroidism or/and hypothyroidism promote AF via perturbation of atrial electrical conduction properties, triggered activity, development of arrhythmogenic fibrosis, or activation of proinflammatory signals. More mechanistic and physio-pharmacological studies will contribute to improve clinical management of patients with thyroid disease and AF.

## 5. Thyroid Disease-Associated Cardiac Comorbidities and AF Risk

Various AF risk factors, including hypertension, diabetes, or myocarditis, are associated with thyroid disorders [77]. Better understanding of how thyroid disease influences the development of inflammation, cardiac fibrosis, and AF in patients with comorbid obesity, myocardial infarction, chronic obstructive pulmonary disease, or heart failure may lead to clarifications on the pathophysiological link between thyroid disease and AF incidence.

### 5.1. Thyroid Hormones, Myocardial Infarction and Cardiac Fibrosis

Myocardial infarction (MI) is an important cause of mortality and morbidity worldwide [78]. MI is cited among the major causes of AF [4]. It has been suggested that serum levels above 0.41 nmol/L of reverse T3 and low levels of T3, may indicate that patients with MI are at high risk of mortality [79]. In a rat model of MI, daily treatment with T3 (5 μg/kg/d) for 8 weeks ameliorated LV contractility, decreased LA fibrosis, normalized LA diameter, and decreased the duration and the incidence (by 88%) of inducible atrial tachyarrhythmia compared to untreated MI-rats. This study also showed that T3 treatment reduced the expression of genes associated with inflammation and oxidative stress while increasing the expression of ion channels involved in the normal contractile machinery [80].

### 5.2. Thyroid Disorders, Hypertension and Chronic Cardiac Hypertrophy

Hypertensive conditions affecting the heart, including systemic hypertension, obesity, chronic obstructive lung disease and pulmonary arterial hypertension (PAH), are associated with an increased risk of AF [81,82,83]. It has been shown that patients with thyrotoxicosis had predominant right heart failure and pulmonary hypertension associated with rapid atrial fibrillation [84,85]. In a Sugen-chronic hypoxia (SuHx) model of PAH in rats, thyroidectomy prevented the development of severe experimental PAH and significantly decreased right ventricular systolic pressure (RVSP) [86]. To our knowledge, experimental studies reporting in vivo data about the impact of thyroid disease or thyroidectomy on lung and systemic-hypertension association with AF are rare. The role of thyroid hormones in the development of pulmonary or hypertension and their potential impact on the development of arrhythmogenic substrate remains unclear.

### 5.3. Thyroid Dysfunction, Inflammation and Myocarditis

Myocarditis is a progressive inflammation of the heart muscle which affects the ability of the heart to pump, leading to severe dilated cardiomyopathy and chronic heart failure [87]. It has been reported that patients with lymphocytic myocarditis showed hyperthyroidism and sinus tachycardia associated with an increased risk of mortality [88]. Animal models of myocarditis and thyroid disease are available using coxsackie-virus to infect susceptible mice. The animals developed autoimmune myocarditis resembling human dilated cardiomyopathy [89]. Experimental data describing the underlying mechanisms of the association between thyroid disease, myocarditis and AF are lacking. It is of interest to understand whether thyroid disease leads to cardiac inflammation and myocarditis, and how this chronic inflammatory status may contribute to atrial arrhythmogenic substrate including atrial fibrosis and conduction abnormalities.

### 5.4. Thyroid Disease, Metabolism Disorder and Atherosclerosis

Evidence suggests that euthyroid state is preferred for the homeostasis of the cardiovascular system [90]. It has been shown that hypothyroidism is associated with atherosclerosis and cardiac ischemic diseases [91]. Hypothyroid disease accelerates atherosclerosis-associated diastolic hypertension, inflammation, high LDL (low-density-lipoprotein cholesterol), diabetes, hypercoagulability and AF [92]. A transgenic model of atherosclerotic mice has shown that atherosclerosis increases the risk of ventricular arrhythmia [93]. The role of thyroid disease in atherosclerosis-induced AF is poorly described, and although animal models seem feasible, no tangible experimental data are available.

## 6. Current Therapy of AF in Thyroid Disease

### 6.1. Atrial Fibrillation and Thyroid Disease: Rhythm Management, Beta-Blockers, Cardioversion

Patients with AF and hypothyroidism should be managed with a rate- vs. rhythm-control as per guideline recommendations. In general, levothyroxine supplementation promptly restores the euthyroid state in hypothyroid patients. Supra-physiological repletion should be suspected in previously well-controlled hypothyroid patients presenting with difficult-to-control AF.

Patients with overt hyperthyroidism present symptoms of adrenergic hyperstimulation such as anxiety, tachycardia, and heat intolerance [75]. Beta-blockers are first-line agents in treating symptomatic patients as they antagonize the hyperthyroid-induced hyperadrenergic state. Selected beta-blockers, such as propranolol, atenolol and metoprolol, also reduce the peripheral conversion of T4 to T3, but the relatively small magnitude of this effect may be of limited clinical significance [94,95]. Atenolol is often preferred because of its daily dosage and extensive experience with this indication [95]. Alternatively, the lowest effective dose of metoprolol should be used during pregnancy [95]. The American Thyroid Association and the American association of Clinical Endocrinologists recommendation that β-adrenergic blockade should be initiated for elderly patients with symptomatic thyrotoxicosis and thyrotoxic patients presenting a resting heart rate greater than 90 bpm or diagnosed with another cardiovascular comorbidity [96].

The choice of medication is crucial and relies on various factors in order to use the most appropriated selective or non-selective beta-blockers, wisely [97]. Propranolol remains one of the mostly used medications in the management of AF patients with thyrotoxicosis, due to its non-selective β-adrenergic antagonism properties [98]. The recommended dose can be as high as 60 to 120 mg orally every 6 hours [98]. To obtain a rapid effect, Propanolol or a selective beta-blocker with a short half-life, esmolol, can be administrated intravenously while monitoring the heart rate [98,99]. Hence, the β-adrenergic blockade should be prescribed with high consideration for patients with hyperthyroid disease and heart failure [99]. The use of other selective beta-blockers like atenolol and metoprolol has also been shown to be efficient in cardiothyroxycosis management [97]. Metoprolol, a β1 cardioselective blocker without agonist activity, was compared favorably with propranolol in controlling symptoms and cardiac complications of hyperthyroidism including tachycardia [100].

It is important to notice that for patients with airway disease including asthma, or chronic obstructive pulmonary disease, treatment with beta-blockers is contraindicated. For rate control, digoxin can be used, but a higher dose than usual is required, because hyperthyroid atrial fibrillation patients are often resistant to digoxin due to increased renal clearance [101].

The anti-thyroid medications (thioamides) methimazole and propylthiouracil (PTU) reduce the production of new thyroid hormone and are used to accelerate recovery to the euthyroid state, while awaiting a definitive therapy (radioiodine ablation, surgery), if indicated. Patients with thyroid storm, an exaggerated and potentially life-threatening form of hyperthyroidism, should be treated in the intensive care unit and may benefit from glucocorticoids, iodinated contrast and/or bile acid sequestrants [98,102] (Table 3).

Hyperthyroidism is a reversible cause of AF. In fact, approximately two-thirds of hyperthyroid patients experience spontaneous return to sinus rhythm after restoration of normal thyroid hormone levels [103,104]. Hence, it is suggested to perform cardioversion only after attempting restoration of the euthyroid state by treating hyperthyroidism and promoting spontaneous reversion to sinus rhythm [103]. In the absence of spontaneous sinus rhythm recovery, cardioversion should be attempted electrically or pharmacologically, but only after the patient has reached euthyroid state [103]. For this reason and because the maintenance of sinus rhythm may be difficult in the setting of uncontrolled hyperthyroidism, a rate-control strategy aiming for a mean heart rate of <100 bpm appears to be a reasonable first-line strategy in most patients. Rate control is usually successful with beta-blockers (first line) and/or non-dihydropyridine calcium channel blockers [15]. Digoxin may be used but is less effective in hyper-adrenergic hyperthyroid patients. In symptomatic patients despite adequate rate control or in patients with persistent AF after restoration of the euthyroid state, a rhythm-control strategy can be utilized after adequate clearance of the left atrial appendage for thrombus, as per guideline recommendations (Table 3). There is no published study on the effectiveness and/or safety of pharmacological antiarrhythmic therapy in hyperthyroid patients, and these drugs should be used with caution, following their respective restrictions.

### 6.2. Amiodarone-Induced Thyrotoxicosis

Amiodarone is one of the most commonly used anti-arrhythmic medications to prevent and treat cardiac arrhythmias including AF [105]. As a class III anti-arrhythmic drug, amiodarone blocks potassium rectifier currents to promote the increase of CM action potential duration and prolongation of the effective refractory period, resulting in the prevention of reentry circuits and attenuation of the risk of AF [106]. Amiodarone contains iodine [107]. Hence, thyrotoxicosis is one of the adverse side-effects often diagnosed among amiodarone users [107]. In clinical practice, AF patients using amiodarone who have hypothyroidism can be treated with levothyroxine, while the treatment of hyperthyroidism often involves corticosteroids, thiamazole, or PTU [106,108]. Amiodarone-induced thyrotoxicosis is challenging to treat, which can result in thyroidectomy, and a higher risk of mortality [108]. Studies have shown that early catheter ablation for paroxysmal AF in patients with amiodarone-induced thyrotoxicosis is a safe and effective approach with similar AF recurrence rate than patients with no history of thyroid disease after 12 months [109,110].

### 6.3. AF Catheter Ablation and Thyroid Disease

Catheter ablation is commonly used in patients with drug-refractory AF and as first-line therapy in selected cases. In patients without a history of thyroid disease, high-normal free T3/T4 (FT3, FT4) levels have been associated with a higher rate of atrial arrhythmia recurrence after AF radiofrequency ablation (hazard ratio 3.31, 95% confidence interval 1.45–7.54; high vs. lowest FT4 quartile) or cryo-ablation (HR 1.187, 95% CI 1.093–1.290) [13,14,111,112,113]. One study found patients on thyroid hormone therapy for hypothyroidism (but with normal TSH levels) to have a higher incidence of non-pulmonary vein AF triggers, particularly right atrial sources, and lower AF-free survival after ablation (64.4% vs. 75.3% for THR vs. no THR; *p* = 0.04) [7]. Similarly, in a study of 477 consecutive patients undergoing first-time AF ablation, Mosishima et al. found hypothyroidism (hazard ratio 3.14) and high-normal TSH (hazard ratio 1.51) to be associated with higher rate of atrial arrhythmia recurrence85 (Table 3). Hence, even with thyroid function markers within the normal range, TSH/FT3/FT4 levels appears to modulate the efficacy of AF catheter ablation.

A larger body of literature has been published on patients with recovered hyperthyroidism undergoing AF ablation. When reported, catheter ablation appears to have the same safety profile in patients with vs. without a history of hyperthyroidism. However, recovered hyperthyroidism appears to be associated with persistent changes to atrial electrophysiology in the form of a higher incidence of non-pulmonary vein ectopy (42% vs. 23%; *p* < 0.01), particularly from the ligament of Marshall, and a higher rate of focal atrial tachycardias vs. controls [109,114]. Moreover, a history of hyperthyroidism has been associated with a higher risk of AF recurrence after ablation (HR 2.07, 95% CI 1.27–3.38) [115] and with the need for more frequent redo procedures compared to controls (35% vs. 7.5%; *p* = 0.01) [116]. The absolute arrhythmia-free survival in this population of mostly paroxysmal patients, acquired with non-invasive monitoring, which tends to overestimate success, has been reported to be in the range of 30–56% with mean follow-ups of 12–15 months, [110,117] which substantially lower than in patients without a history of thyroid disease. Others have reported no association between treated hyperthyroidism and long-term AF-free survival [118,119]. Finally, in a preliminary report of 39 patients with active overt or subclinical hyperthyroidism, 60% of patients experience atrial arrhythmia recurrence at 18 months, which was significantly higher than controls, despite extensive ablation (cavo-tricuspid isthmus line, pulmonary vein isolation, lateral mitral isthmus line and roof line) [120] (Table 3).

### 6.4. Thyroid Disease, Stroke, and Atrial Fibrillation: Relevance of Anticoagulation Therapy

Thyroid disease has been associated with a range of biochemical abnormalities in coagulation and fibrinolysis. In particular, hyperthyroid patients have been found to have elevated plasma fibrinogen, von Willebrand factor and factor X levels, among others [121,122,123,124]. The clinical significance of these laboratory findings is not entirely clear; a limited body of evidence appears to suggest an association between hyperthyroidism and an increased risk of venous and arterial thrombosis [125]. More recent work found the increased risk of stroke in hyperthyroid patients to be accounted for by the increased prevalence of AF in this population, re-igniting the debate on whether hyperthyroidism itself is responsible for the incremental risk or whether the risk is mediated by AF [125]. Conversely, hypothyroid patients appear to be at higher risk of bleeding complications [126]. However, the incremental risk is insufficient to impact the indications for oral anticoagulation therapy (Table 3).

**Table 3 cells-11-04047-t003:** Available strategies in the management of patients affected by thyroid disease and AF. Patients with both thyroid disorders and AF are treated with caution to control heart rhythm, prevent blood clots while treating hormonal deregulation and avoid toxic adverse medication-interactions.

Anti Thyroid & AF Treatment	Medication or Surgery	Main Outcomes	Refs.
AF Rhythm management and Thyroid disease	Beta-blockers	Propranolol, atenolol and metoprolol to control tachycardia & also to reduce the peripheral conversion of T4 to T3.	[94,95]
	Thionamides	Inhibit thyroid hormone production to accelerate euthyroid state till definitive treatment is available.	[98,102]
	Glycocorticoids	Cardiac rhythm reverted to SR in 86% patients.	[127]
	non-dihydropyridine calcium channel blockers	Rate control in combination with β-blockers	[15]
AF catheter ablation and Thyroid disease	AF radiofrequency ablation	Higher T3/T4 levels correlated with higher rate of atrial arrhythmia recurrence after AF radiofrequency ablation.	[13,14,111,112]
		Hypothyroidism & high normal TSH level corelated with AF occurence	[113]
Prevent Stroke, in AF and Thyroid disease	Apixaban	Effectively prevented stroke in patients with non-valvular AF and hypo- or hyperthyroidism	[125,126]
	Warfarin	CCS/CHRS recommendsuse of warfarin in hyperthorid patients will euthyroid state is reached.	[15]

Patients with AF and thyroid disease should be evaluated using a guideline-recommended stroke risk-stratification score. It is however unclear if overt hyperthyroidism in and of itself constitutes an indication for anticoagulation. The Canadian Cardiovascular Society/Canadian Heart Rhythm Society favors anticoagulating hyperthyroid patients with warfarin until the euthyroid state is restored [15]. The American Heart Association/American College of Cardiology/Heart Rhythm Society as well as the European Society of Cardiology guidelines recommend following the CHA2DS2-VASc risk score and therefore not anticoagulating patients solely on the basis of hyperthyroidism [123,124]. In a sub study of the Apixaban for Reduction in Stroke and Other Thromboembolic Events in Atrial Fibrillation (ARISTOTLE) trial comparing apixaban vs. warfarin, apixaban was found to be equally effective at preventing stroke in patients with non-valvular AF and hypo- or hyperthyroidism vs. no thyroid disease [125]. A more recent Taiwanese cohort study found the direct oral anticoagulants apixaban, dabigatran and rivaroxaban to be safe and effective at stroke prevention in patients with AF and hyperthyroidism [126,127]. Hence, direct oral anticoagulants appear safe/effective in patients with AF and thyroid disease (Table 3).

## 7. Conclusions

Recent clinical evidence shows that hyperthyroidism is associated with a significant risk of atrial fibrillation. Experimental studies at the cellular level and in animal models of thyroid disease have shown that thyroid hormones mediate cardiac structural, functional, and electrical remodeling. The atrial myocardium is highly sensitive to thyroid hormones that influence atria physiology in an endocrine and paracrine manner to promote the development of atrial fibrosis, which increases AF susceptibility. Strategies targeting thyroid hormones and their receptors are potential therapeutic approaches to combat thyroid disease-associated arrhythmogenic substrate.

## 8. Highlights

Recent clinical reports suggest that hyperthyroidism is associated with increased AF risk.Experimental investigations revealed that hyperthyroidism and hypothyroidism provoke atrial electrical conduction slowing and fibrosis, promoting AF substrate.Thyroid hormones interact with specific receptors expressed on atrial cardiomyocytes and fibroblasts to activate intracellular arrhythmogenic signaling.Therapy includes careful interaction of antiarrhythmic and anti-thyroid-disease medications.

## Figures and Tables

**Figure 1 cells-11-04047-f001:**
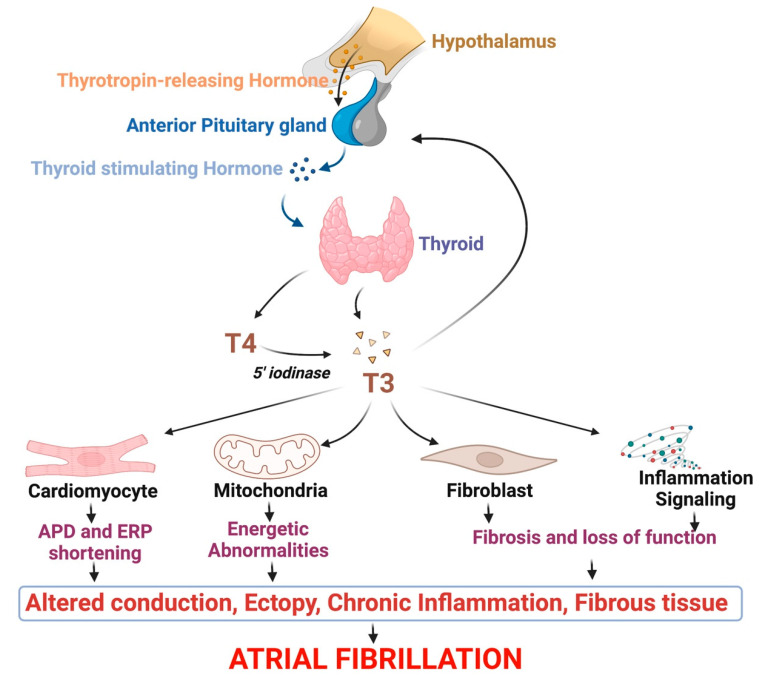
A schematic cascade of events from hypothalamus activity and thyroid malfunction to promotion of AF. Pre-clinical and clinical evidence show that hyperthyroidism, and eventually hypothyroidism, are responsible for structural and functional changes affecting cardiac cells, including atrial cardiomyocytes and atrial fibroblasts, leading to the development of arrhythmogenic substrate characterized by electrical conduction slowing, mitochondrial energetic perturbation, and inflammation-induced fibrosis and cardiotoxicity promoting occurrence and maintenance of atrial fibrillation.

**Figure 2 cells-11-04047-f002:**
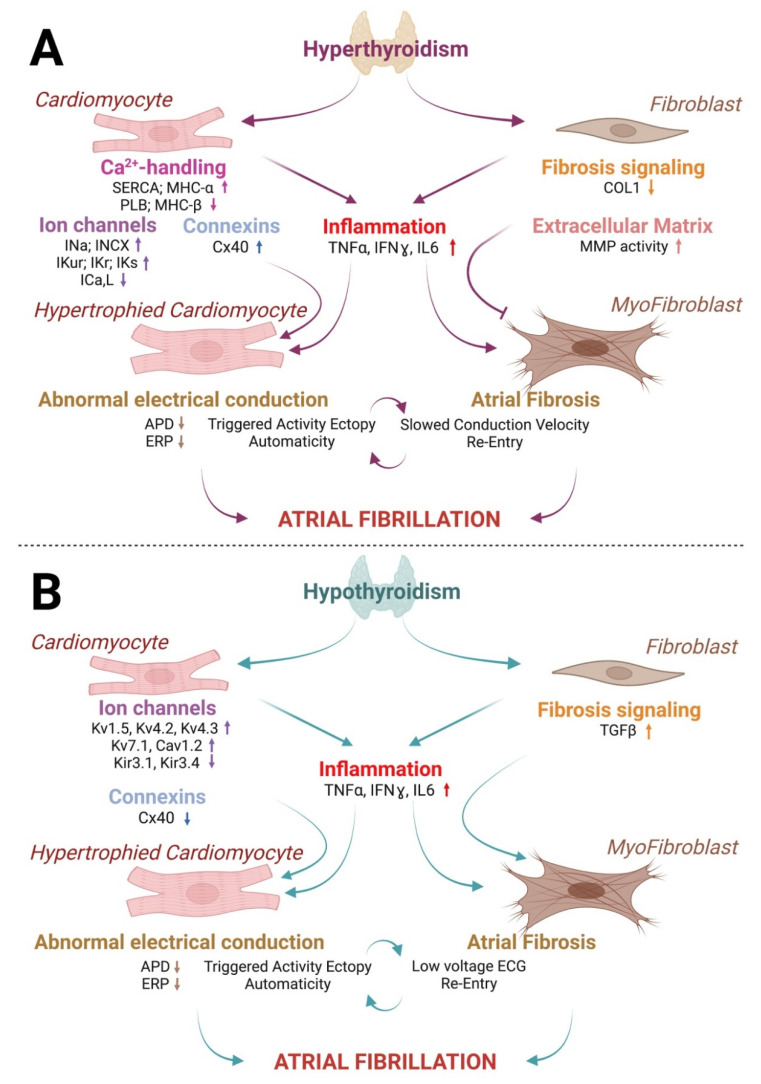
AF-promoting CM and FB remodeling, induced by hyperthyroidism compared to hypothyroidism. (**A**) Hyperthyroidism may promote AF via the development of abnormal electrical conduction, ectopy, and re-entry essentially induced by CM remodeling including perturbed calcium handling, increased inward sodium currents and decreased L-type calcium channel expression. (**B**) Hypothyroidism-associated vulnerability to arrhythmias is generated via myocardial remodeling and essentially FB-induced fibrosis leading to slowed atrial conduction and increased incidence of AF.

**Table 1 cells-11-04047-t001:** Clinical evidence of arrhythmogenicity in patients with thyroid disease. Clinical trials and reports assessing the prevalence or/and incidence of atrial fibrillation in patients with thyroid disorders.

Thyroidism	Demography	Study Duration	Association of Thyroid Disease with AF Incidence	References
Low TSH	Population: 726 patients (273 women, 453 men) Age: 65.5 ± 13 years old	1.7 years	HR: 1.67 95% CI; 1.7–14.0; *p* ≤ 0.05	[17]
Low TSH	Population: 2007 patients (1193 women; 814 men). Age: ≥ 60 years old	10 years	HR: 3.1 95% CI; 1.7–5.5; *p* < 0.001	[12]
High Free T4	Population: 1426 patients (841 women, 585 men) Age: 68 ± 8 years old	8 years	HR: 1.62 95% CI; 0.84–3.14; *p* < 0.06	[4]
High Free T4	Population: 1095 patients (298 women; 797 men) Age: 60 ± 10 years old	3 years	HR: 1.15 95% CI; 1.03–1.29; *p* = 0.014	[13]
High Free T4	Population: 244 patients (67 women; 177 men) Age: 55 ± 12 years old	1 year	HR: 3.31 95% CI; 1.45–7.54; *p* = 0.004	[14]
High Free T4	Population: 10,318 patients (5886 women, 4432 men) Age: 65 ± 10 years old	9 years	HR: 2.28 95% CI; 1.31–3.97; *p* ≤ 0.05	[20]
High Free T4	Population: 174,914 patients (113,519 women, 61,395 men) Age: 65 ± 10 years old	7 years	HR: 1.16 95% CI; 1.10–1.21; *p* < 0.0001	[21]
High Free TA	Population: 30,085 patients (15,584 women, 159,390 men) Age: ~69 years old	17 years	HR: 1.45 95% CI; 1.26–1.66; *p* ≤ 0.001	[22]

## Data Availability

Not applicable.

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
