# Peer review of "Mechanisms and Management of Thyroid Disease and Atrial Fibrillation: Impact of Atrial Electrical Remodeling and Cardiac Fibrosis"

_cells, 2022, doi:10.3390/cells11244047_

Round 1
Reviewer 1 Report
This study highlights thyroid disease-induced arrhythmogenic remodeling on atrial myocardium.The study therefore summarizes clinical data and tries to explain this findings by discussing experimental data.
The review is well written. The key messages are illustrated in comprehensive figures.
Minor comments:
line 203: 66 is no highlighted as a reference
Author Response
We warmfly tham the reviewer for his/her appreciation of our article.
The reference is now corrected in the current version of the manuscript.

Reviewer 2 Report
Dear Sir/Madam,
I had the opportunity to act as a reviewer on the recent submission by Takawale et al. to the Cells journal.
The authors review the mechanisms and management of atrial fibrillation in thyroid disease. Very interesting is the overview of the link between the hypothyroidism and atrial fibrillation.
The manuscript is very well structured and written. However, some issues need to be addressed:
1. I recommend adding an additional section with focus on amiodarone induced thyrotoxicosis and its management: this is highly relevant in daily practice.
2. The authors refer to the management of atrial fibrillation in thyrotoxicosis: I recomment commenting on the differences between non-selective beta-blockers and beta 1 selective beta-blockers.
3. The authors state that hyperthyroidism is a reversible cause for atrial fibrillation: does this mean that after restoring sinus rhythm the patients should not be anticoagulated irrespective of the CHA2DS2VASc score? Please comment on this.
Best regards,
Author Response
REVIEWER #2
Dear Sir/Madam,
I had the opportunity to act as a reviewer on the recent submission by Takawale et al. to the Cells journal.
The authors review the mechanisms and management of atrial fibrillation in thyroid disease. Very interesting is the overview of the link between the hypothyroidism and atrial fibrillation.
The manuscript is very well structured and written. However, some issues need to be addressed:
- I recommend adding an additional section with focus on amiodarone induced thyrotoxicosis and its management: this is highly relevant in daily practice.
We thank the reviewer for this relevant comment. A paragraph is now added to discuss the association between thyrotoxicosis and amiodarone, one of the medications involved in AF treatment. This is now added in the section 6.2. titled: Amiodarone-induced thyrotoxicosis. The reference list has been completed accordingly.
<< Amiodarone is one of the most commonly used anti-arrhythmic medications to prevent and treat cardiac arrhythmias including AF [105]. As a class III anti-arrhythmic drug, amiodarone blocks potassium rectifier currents to promote the increase of CM action po-tential duration and prolongation of the effective refractory period, resulting in the preven-tion of reentry circuits and attenuation of the risk of AF [106]. Amiodarone contains iodine [107]. Hence, thyrotoxicosis is one of the adverse side-effects often diagnosed among amiodarone users [107]. In clinical practice, AF patients using amiodarone who have hy-pothyroidism can be treated with levothyroxine, while the treatment of hyperthyroidism often involves corticosteroids, thiamazole, or PTU [106,108]. Amiodarone-induced thyro-toxicosis is challenging to treat, which can result in thyroidectomy, and a higher risk of mortality [108]. Studies have shown that early catheter ablation for paroxysmal AF in pa-tients with amiodarone-induced thyrotoxicosis is a safe and effective approach with sim-ilar AF recurrence rate than patients with no history of thyroid disease after 12 months [109,110].>>
- The authors refer to the management of atrial fibrillation in thyrotoxicosis: I recommend commenting on the differences between non-selective beta-blockers and beta 1 selective beta-blockers.
Thank you very much. We now provide a more extensive discussion of the non-selective and selective beta-blockers utilization in the management of AF in thyrotoxicosis. This is now readable in section 6.1, titled: 6.1. Atrial Fibrillation and Thyroid Disease: Rhythm Management, beta-blockers, cardioversion. The reference list has been corrected accordingly.
<< Beta-blockers are first-line agents in treating symptomatic patients as they antagonize the hyperthyroid-induced hyperadrenergic state. Selected beta-blockers, such as propranolol, atenolol and metoprolol, also reduce the peripheral conversion of T4 to T3, but the relatively small magnitude of this effect may be of limited clinical significance [94,95]. Atenolol is often preferred because of its daily dosage and extensive experience with this indication [95]. Alternatively, the lowest effective dose of metoprolol should be used during pregnancy [95]. The American Thyroid Association and the American association of Clinical Endocrinologists recommendation that β-adrenergic blockade should be initiated for elderly patients with symptomatic thyrotoxicosis and thyrotoxic patients presenting a resting heart rate greater than 90 bpm or diagnosed with another cardiovascular comorbidity [96].
The choice of medication is crucial and relies on various factors in order to use the most appropriated selective or non-selective beta-blockers, wisely [97]. Propranolol remains one of the mostly used medications in the management of AF patients with thyrotoxicosis, due to its non-selective β-adrenergic antagonism properties [98]. The recommended dose can be as high as 60 to 120 mg orally every 6 hours [98]. To obtain a rapid effect, Propanolol or a selective beta-blocker with a short half-life, esmolol, can be administrated intravenously while monitoring the heart rate [98,99]. Hence, the β-adrenergic blockade should be prescribed with high consideration for patients with hyperthyroid disease and heart failure [99]. The use of other selective beta-blockers like atenolol and metoprolol has also been shown to be efficient in cardiothyroxycosis management [97]. Metoprolol, a β1 cardioselective blocker without agonist activity, was compared favorably with propranolol in controlling symptoms and cardiac complications of hyperthyroidism including tachycardia [100].
It is important to notice that for patients with airway disease including asthma, or chronic obstructive pulmonary disease, treatment with beta-blockers is contraindicated. For rate control, digoxin can be used, but a higher dose than usual is required, because hyperthyroid atrial fibrillation patient are often resistant to digoxin due to increased renal clearance [101]. >>
- The authors state that hyperthyroidism is a reversible cause for atrial fibrillation: does this mean that after restoring sinus rhythm, the patients should not be anticoagulated irrespective of the CHA2DS2VASc score? Please comment on this.
Although we support that hyperthyroidism is a reversible cause of AF, we provide a discussion about the controversy regarding the use of anticoagulation drugs. We addressed the recommendations from The Canadian Cardiovascular Society/Canadian Heart Rhythm Society; The American Heart Association/American College of Cardiology/Heart Rhythm Society; and the European Society of Cardiology. Moreover, we report data from the ARISTOLE trial and a Taiwanese trial assessing the safety of anticoagulation in patients with AF and thyroid disease.
This information is available in the second part of the current section 6.4., which is titled: 6.4. Thyroid Disease, Stroke, and Atrial Fibrillation: Relevance of anticoagulation therapy.
<< Patients with AF and thyroid disease should be evaluated using a guideline-recommended stroke risk-stratification score. It is however unclear if overt hyperthyroidism in and of itself constitutes an indication for anticoagulation. The Canadian Cardiovascular Society/Canadian Heart Rhythm Society favors anticoagulating hyperthyroid patients with warfarin until the euthyroid state is restored [15]. The American Heart Association/American College of Cardiology/Heart Rhythm Society as well as the European Society of Cardiology guidelines recommend following the CHA2DS2-VASc risk score and therefore not anticoagulating patients solely on the basis of hyperthyroidism [123,124]. In a sub study of the Apixaban for Reduction in Stroke and Other Thromboembolic Events in Atrial Fibrillation (ARISTOTLE) trial comparing apixaban vs warfarin, apixaban was found to be equally effective at preventing stroke in patients with non-valvular AF and hypo- or hyperthyroidism vs no thyroid disease [125]. A more recent Taiwanese cohort study found the direct oral anticoagulants apixaban, dabigatran and rivaroxaban to be safe and effective at stroke prevention in patients with AF and hyperthyroidism [126,127]. Hence, direct oral anticoagulants appear safe/effective in patients with AF and thyroid disease (Table 3). >>

Reviewer 3 Report
This is an interesting review, focused on the experimental and clinical evidence linking atrial fibrosis, Atrial fibrillation and thyroid diseases;
My very few concerns that can be arranged with 1-3 paragraphs for each are
1) What about acute onset of atrial fibrillation following thyroid hormones imbalance
2) What about electrical Cardioversion and Anticoagulation therapy and
3) What about Thyroid hormones and comorbidities such as hypertension, Diabetes etc
Author Response
This is an interesting review, focused on the experimental and clinical evidence linking atrial fibrosis, Atrial fibrillation and thyroid diseases;
My very few concerns that can be arranged with 1-3 paragraphs for each are
1) What about acute onset of atrial fibrillation following thyroid hormones imbalance.
We are thankful to the reviewer for he/her constructive comments and suggestions.
We provide clinical data and discussed the association between new-onset AF and thyroid disease in the first part of section 2, titled: Clinical evidence of thyroid disease and AF.
<< There is a well-established epidemiological association between thyroid disease and AF such that thyroid function testing is recommended in all patients with new-onset AF [15]. Overt hyperthyroidism (low TSH, elevated T4) has the strongest association with AF. It increases the risk of AF up to 3-fold [12,16], but is relatively uncommon [17]. More recently, subclinical hyperthyroidism (low TSH with normal T3/T4), often regarded as a benign entity, has also been reported to carry an increased risk of AF (hazard ratio in the range of 1.5) [4,18-22]. Similarly, a large phenome-wide association study found TSH levels within the normal range to be inversely related to AF (odds ratio 0.91; 95% confidence interval, 0.83-0.99; p = 0.04), supporting the notion that even subclinical increases in thyroid function are associated with AF [23]. Subclinical hyperthyroidism is present in up to 13% of patients with new-onset AF such that it may represent an underappreciated reversible cause or contributing factor in a sizeable number of patients [24].
>>
2) What about electrical Cardioversion and Anticoagulation therapy
We developed the discussion about the importance of restoring sinus rhythm while considering the prior attempt of restoring euthyroid state. This is described in section 6.1. titled: 6.1. Atrial Fibrillation Rhythm Management and Thyroid Disease.
<<… Hence, it is suggested to perform cardioversion only after attempting restoration of the euthyroid state by treating hyperthyroidism and promoting spontaneous reversion to sinus rhythm [103]. In the absence of spontaneous sinus rhythm recovery, cardioversion should be attempted electrically or pharmacologically, but only after the patient has reached euthyroid state [103]. >>
As also suggested by reviewer 1, here we provide a discussion about the use of anticoagulation drugs. We addressed the guidelines from The Canadian Cardiovascular Society/Canadian Heart Rhythm Society; The American Heart Association/American College of Cardiology/Heart Rhythm Society; and the European Society of Cardiology. In addition, we report information from the ARISTOLE trial assessing the safety of anticoagulation use in patients with AF and thyroid disease.
This information is available in the second part of the current section 6.4., which is titled: Thyroid Disease, Stroke, and Atrial Fibrillation: Relevance of anticoagulation therapy.
<< Patients with AF and thyroid disease should be evaluated using a guideline-recommended stroke risk-stratification score. It is however unclear if overt hyperthyroidism in and of itself constitutes an indication for anticoagulation. The Canadian Cardiovascular Society/Canadian Heart Rhythm Society favors anticoagulating hyperthyroid patients with warfarin until the euthyroid state is restored [15]. The American Heart Association/American College of Cardiology/Heart Rhythm Society as well as the European Society of Cardiology guidelines recommend following the CHA2DS2-VASc risk score and therefore not anticoagulating patients solely on the basis of hyperthyroidism [123,124]. In a sub study of the Apixaban for Reduction in Stroke and Other Thromboembolic Events in Atrial Fibrillation (ARISTOTLE) trial comparing apixaban vs warfarin, apixaban was found to be equally effective at preventing stroke in patients with non-valvular AF and hypo- or hyperthyroidism vs no thyroid disease [125]. A more recent Taiwanese cohort study found the direct oral anticoagulants apixaban, dabigatran and rivaroxaban to be safe and effective at stroke prevention in patients with AF and hyperthyroidism [126,127]. Hence, direct oral anticoagulants appear safe/effective in patients with AF and thyroid disease (Table 3).>>
3) What about Thyroid hormones and comorbidities such as hypertension, Diabetes etc.
The reviewers’ suggestion is very interesting. Although we believe that such discussion might be beyond the scope of this review, as we want to focus on thyroid disease, cardiac fibrosis and AF per se, we therefore, provide a brief assessment of the known evidence of the association between thyroid disorders and cardiac comorbidities described as important AF risk factors, including myocardial infarction and hypertension-induced cardiac remodeling.
This is now written in section 5. The reference list has been corrected accordingly.
<< 5. Thyroid disease-associated cardiac comorbidities and AF risk.
Various AF risk factors, including hypertension, diabetes, or myocarditis, are associated with thyroid disorders [77]. Better understanding of how thyroid disease influences the development of inflammation, cardiac fibrosis, and AF in patients with comorbid obesity, myocardial infarction, chronic obstructive pulmonary disease, or heart failure may lead to clarifications on the pathophysiological link between thyroid disease and AF incidence.
5.1. Thyroid hormones, Myocardial Infarction and cardiac fibrosis.
Myocardial infarction (MI) is an important cause of mortality and morbidity worldwide [78]. MI is cited among the major causes of AF [4]. It has been suggested that serum levels above 0.41 nmol/L of reverse T3 and low levels of T3, may indicate that patients with MI are at high risk of mortality [79]. In a rat model of MI, daily treatment with T3 (5 µg/kg/d) for 8 weeks ameliorated LV contractility, decreased LA fibrosis, normalized LA diameter, and decreased the duration and the incidence (by 88%) of inducible atrial tachyarrhythmia compared to untreated MI-rats. This study also showed that T3 treatment reduced the expression of genes associated with inflammation and oxidative stress while increasing the expression of ion channels involved in the normal contractile machinery [80].
5.2. Thyroid disorders, hypertension and chronic cardiac hypertrophy.
Hypertensive conditions affecting the heart, including systemic hypertension, obesity, chronic obstructive lung disease and pulmonary arterial hypertension (PAH), are associated with an increased risk of AF [81,82,83]. It has been shown that patients with thyrotoxicosis had predominant right heart failure and pulmonary hypertension associated with rapid atrial fibrillation [84,85]. In a Sugen-chronic hypoxia (SuHx) model of PAH in rats, thyroidectomy prevented the development of severe experimental PAH and significantly decreased right ventricular systolic pressure (RVSP) [86]. To our knowledge, experimental studies reporting in vivo data about the impact of thyroid disease or thyroidectomy on lung and systemic-hypertension association with AF are rare. The role of thyroid hormones in the development of pulmonary or hypertension and their potential impact on the development of arrhythmogenic substrate remains unclear.
5.3. Thyroid dysfunction, inflammation and myocarditis.
Myocarditis is a progressive inflammation of the heart muscle which affects the ability of the heart to pump, leading to severe dilated cardiomyopathy and chronic heart failure [87]. It has been reported that patients with lymphocytic myocarditis showed hyperthyroidism and sinus tachycardia associated with an increased risk of mortality [88]. Animal models of myocarditis and thyroid disease are available using coxsackie-virus to infect susceptible mice. The animals developed autoimmune myocarditis resembling human dilated cardiomyopathy [89]. Experimental data describing the underlying mechanisms of the association between thyroid disease, myocarditis and AF are lacking. It is of interest to understand whether thyroid disease leads to cardiac inflammation and myocarditis, and how this chronic inflammatory status may contribute to atrial arrhythmogenic substrate including atrial fibrosis and conduction abnormalities.
5.4. Thyroid disease, metabolism disorder and atherosclerosis.
Evidence suggests that euthyroid state is preferred for the homeostasis of the cardiovascular system [90]. It has been shown that hypothyroidism is associated with atherosclerosis and cardiac ischemic diseases [91]. Hypothyroid disease accelerates atherosclerosis-associated diastolic hypertension, inflammation, high LDL (low-density-lipoprotein cholesterol), diabetes, hypercoagulability and AF [92]. A transgenic model of atherosclerotic mice has shown that atherosclerosis increases the risk of ventricular arrhythmia [93]. The role of thyroid disease in atherosclerosis-induced AF is poorly described, and although animal models seem feasible, no tangible experimental data are available. >>

Round 2
Reviewer 2 Report
Dear Sir/Madam,
Thank you for reviewing the manuscript and addressing the mentioned issues. These were adequately answered. Therefore, the manuscript seems suitable for publishing in the present form.
Best regards